# Fabrication of an Electrochemical Aptasensor Composed of Multifunctional DNA Three-Way Junction on Au Microgap Electrode for Interferon Gamma Detection in Human Serum

**DOI:** 10.3390/biomedicines9060692

**Published:** 2021-06-18

**Authors:** Seungwoo Noh, Jinmyeong Kim, Chulhwan Park, Junhong Min, Taek Lee

**Affiliations:** 1Department of Chemical Engineering, Kwangwoon University, Seoul 01897, Korea; nsw26510@naver.com (S.N.); wls629@icloud.com (J.K.); chpark@kw.ac.kr (C.P.); 2School of Integrative Engineering, Chung-Ang University, Seoul 06974, Korea

**Keywords:** aptasensor, biomarker, IFN-γ detection, electrochemical biosensor

## Abstract

Interferon gamma (IFN-γ) is an important cytokine with antiviral, antibacterial, and immunosuppressive properties. It has been used as a biomarker for the early detection of several diseases, including cancer, human immunodeficiency virus (HIV), tuberculosis, and paratuberculosis. In this study, we developed an electrochemical biosensor composed of multifunctional DNA 3WJ to detect IFN-γ level with high sensitivity. Each multifunctional triple-stranded aptamer (MF-3WJ) was designed to have an IFN-γ aptamer sequence, anchoring region (thiol group), and 4C–C (cytosine–cytosine) mismatch sequence (signal generation), which could introduce silver ions. To generate the electrochemical signal, four Ag^+^ ions were intercalated (3wj b-3wj c) in the 4C–C mismatch sequence. MF-3WJ was assembled through the annealing step, and the assembly of MF-3WJ was confirmed by 8% tris–boric–EDTA native polyacrylamide gel electrophoresis. The Au microgap electrode was manufactured to load sample volumes of 5 µL. The reliability of electrochemical biosensor measurement was established by enabling the measurement of seven samples from one Au microgap electrode. MF-3WJ was immobilized on the Au microgap electrode. Then, cyclic voltammetry and electrochemical impedance spectroscopy were performed to confirm the electrochemical properties of MF-3WJ. To test the electrochemical biosensor’s ability to detect IFN-γ, the limit of detection (LOD) and selectivity tests were performed by square wave voltammetry. A linear region was observed in the concentration range of 1 pg/mL–10 ng/mL of IFN-γ. The LOD of the fabricated electrochemical biosensor was 0.67 pg/mL. In addition, for the clinical test, the LOD test was carried out for IFN-γ diluted in 10% human serum samples in the concentration range of 1 pg/mL–10 ng/mL, and the LOD was obtained at 0.42 pg/mL.

## 1. Introduction

Tuberculosis (TB) is an infectious disease caused by several types of *Mycobacterium*. Although TB is as old as human history, it remains an unresolved problem that threatens public health [1,2]. According to the World Health Organization (WHO), in 2019 alone, 1.4 million people died from TB, and about 10 million people were infected with TB [3,4]. The number of people infected with TB is declining by about 2% per year, but the number of new patients is still increasing due to the growing population [5]. The most effective way to manage TB risk is to quickly diagnose active TB patients and provide appropriate treatment to prevent the spread of TB bacteria, thereby preventing the occurrence of new TB patients.

For TB diagnosis, various methods, such as chest radiography [6], *Mycobacterium tuberculosis* smear [7], *M. tuberculosis* molecular diagnosis [8], susceptibility to isoniazid and rifampin [9], and tuberculin test (TST), have been carried out [10]. TST, which is used to investigate infection with *M. tuberculosis*, may produce false-positive results due to the Bacillus Calmette–Guérin vaccine. The interferon gamma (IFN-γ) test was introduced to address this issue [11,12]. The IFN-γ test involves an antigen that stimulates T cells [13] sensitized by *M. tuberculosis*, and is used to check for infection with *M. tuberculosis* through an enzyme-linked immunosorbent assay (ELISA) [14,15]. Apart from resolving the false-positive problem of TST, the IFN-γ test is in the spotlight as a test method that can be used to diagnose latent tuberculosis. While the ELISA-based technique is a commonly used method capable of quantifying IFN-γ in a sensitive and specific manner, production of an antibody is time-consuming and costly. Other disadvantages associated with this method include several washing steps, temperature sensitivity, and difficulty in miniaturization and multiplexing owing to the use of several reagents. An aptamer is a single-stranded nucleic acid that has many advantages over antibodies, such as temperature stability and low manufacturing cost. It is composed of a relatively simple nucleic acid sequence that is easy to modify and has excellent convenience and flexibility in structural design [16,17,18]. In this study, we attempted to insert a redox label for electrochemical detection into an aptamer by utilizing its excellent flexibility.

Electrochemical biosensor [19] is a powerful analysis tool due to its portability, fast response time, and high sensitivity. The miniaturization of modern microelectronics is made possible by constructing microelectrodes that are suitable for detecting small samples [20,21]. The large-scale production and low cost of electronic devices are important reasons that make the electrochemical approach more attractive for high-throughput analysis.

As shown in Figure 1, the detection strategy proposed in this study involved the self-assembly of a multifunctional triple-stranded aptamer (MF-3WJ) on a Au microgap electrode. The change in redox current was quantified using square wave voltammetry (SWV), and it was confirmed that this change was dependent on the concentration of IFN-γ. MF-3WJ consisted of three arms, namely, an IFN-γ aptamer sequence, a thiol group, and a C–C mismatch sequence. Ag^+^ was inserted between the C–C mismatched sequences to stabilize the double strand, and Ag^+^ was used as a redox label [22]. The reliability of the electrochemical biosensor was secured by measuring seven samples on one Au microgap electrode designed to minimize the use of IFN-γ samples.

## 2. Experimental Details

### 2.1. Materials

IFN-γ (16.7 kDa), C-reactive protein (CRP, 23 kDa), and tumor necrosis factor-alpha (TNF-α, 17.4 kDa) were purchased from Sino Biological Inc. (Beijing, China) and kept frozen at −20 °C. Myoglobin, hemoglobin, bovine serum albumin (BSA), potassium hexacyanoferrate (III), and potassium hexacyanoferrate (II) trihydrate were purchased from Sigma-Aldrich (Saint Louis, MO, USA). Silver(Ⅰ) nitrate (AgNO_3_) and 4-(2-hydroxyethyl)-1-piperazineethanesulfonic acid (HEPES) were purchased from Daejung (Gyeonggi-do, South Korea). IFN-apt-3wj-a was synthesized by GenoTech (Daejeon, South Korea), and 3wj-b and SH-3wj-c were synthesized by Integrated Device Technology, Inc. (San Jose, CA, USA). The sequence of IFN-apt-3wj-a was 5′-CCG CCC AAA TCC CTA AGA GAA GAC TGT AAT GAC ATC AAA CCA GAC ACA CTA CAC ACG CAT TGC CAT GTG TAT GTG GG-3′ (77mer). The sequence of 3wj-b was 5′-TTC ACC CCT GAC ATG GCA A-3′ (19mer), and that of SH-3wj-c was 5′-CCC ACA TAC CAC CCC TGA A-3′ (19mer). IFN-apt sequence was followed by [23]. All oligonucleotides were diluted in nuclease-free water and kept frozen at −20 °C. Mini-Protean Tetra Cell and PowerPac^™^ power supplies were purchased from Bio-Rad (Hercules, CA, USA) and were used for gel electrophoresis to analyze DNA. For the clinical test, human serum used extracts from human male AB plasma (Sigma). Human serum was mixed with various diluted IFN samples with phosphate-buffered saline (PBS).

### 2.2. Assembly of the Multifunctional DNA 3WJ Structure

The MF-3WJ structure consisted of three fragments with each functioning as IFN-γ aptamer sequence, anchoring region using thiol functional group, and electrochemical signal reporter through C–C mismatch sequence. Ag^+^ was inserted into the C–C mismatch sequence and used as an oxidation–reduction label [24]. MF-3WJ was assembled in TMS buffer (50 mM tris, 10 mM MgCl_2_, 100 mM NaCl) by annealing. The annealing process included heating the same molar ratio ssDNA at 80 °C for 5 min and then cooling to 4 °C at a rate of 1 °C/min. The assembled MF-3WJ structure was confirmed using tris–boric–EDTA native polyacrylamide gel electrophoresis (TBE native PAGE). The electrophoresis status was examined using 100 bp DNA ladder (Bioneer Inc., Daejeon, South Korea). Each DNA sample was electrophoresed at 70 V for 60 min [25].

### 2.3. Fabrication/Preparation of the Au Microgap Electrode

The Au microgap electrode [26] was designed by the Korea Advanced Nano Fab Center (Daejon, Korea). Metal deposition of Cr (2 nm) and Au (50 nm) on the SiO_2_ was carried out using an electron beam evaporator. The Au microgap electrode was 10 mm wide and 15 mm long and had an area of 150 mm^2^. The working electrode area was 0.09 mm^2^, and the gap between the working electrodes and the counter electrode was 10 µm. For pretreatment of the electrodes, the electrodes were performed for 15 min with ultrasonication in acetone solution. Then, the electrode was sequentially washed with ethanol and distilled water and dried using nitrogen gas. For electrochemical measurements, polydimethylsiloxane (PDMS) was attached to the electrode and used as a chamber. PDMS was prepared by reacting sylgard 184A (DOW, Midland, MI, USA) and sylgard 184B (DOW, Midland, MI, USA) in a 10:1 ratio and drying in an oven at 70 °C for 1.5 h. Figure 2 shows the electrode drawing and preparation process [27].

### 2.4. Electrochemical Analysis

All electrochemical experiments were performed on a three-electrode system using an electrochemical workstation (CHI 760E, CH Instruments, Austin, TX, USA). The three-electrode system consisted of a Ag/AgCl reference electrode (CH Instruments, USA) and a Au microgap electrode as the counter electrode and working electrode. Cyclic voltammetry (CV) was performed in 10 mM HEPES (pH 7.03) and 5 mM [Fe (CN)_6_]^3−/4−^ at a scan rate of 0.03 V/s and a voltage range of −0.3 to 0.6 V [28]. Electrochemical impedance spectroscopy (EIS) was performed with a voltage of 0.25 V in the frequency range of 1 Hz to 100 kHz in the same buffer. Square wave voltammetry (SWV) was performed in 10 mM HEPES (pH 7.03) in the potential range of −0.3 to 0.6 V and at a frequency of 15 Hz and an amplitude of 0.025 V.

## 3. Results and Discussion

### 3.1. Multifunctional DNA 3WJ Feasibility Evaluation

Multifunctional DNA consisted of three different ssDNA sequences. IFN-γ-Apt-3wj-a was an aptamer sequence that specifically bound to IFN-γ, SH-3wj-c was modified with a thiol group to immobilize on the substrate, and 3wj-b/3wj-c had four consecutive cytosine sequences to introduce Ag^+^. Ag^+^ stabilized the duplex DNA strand and was used as an oxidation–reduction label. Figure 3A shows the expected two-dimensional structure of MF-3WJ. The assembly of the MF-3WJ structure was confirmed using 8% TBE PAGE (Figure 3B) [25]. The bands in lanes 2, 3, and 4 showed single-stranded DNA of 3wj-a, 3wj-b, and 3wj-c, respectively. Lane 5 showed an assembled MF-3WJ structural band. In lane 5, a new band that was not observed in lanes 2–4 formed on top, indicating that MF-3WJ was assembled.

### 3.2. Comparison of the Electrochemical Performances of the Au Microgap Electrode and Au Substrate

MF-3WJ was immobilized to the prepared Au microgap electrode using a self-assembly process (gold–sulfur bond). The electrochemical capability of the prepared Au microgap electrodes was compared with a Au substrate (2 cm × 0.8 cm, National NanoFab Center, Daejeon, South Korea). Au substrate samples were prepared in the same way as Au microgap electrodes. The electrochemical performance was evaluated through CV and EIS. In the presence of [Fe(CN)_6_]^3−4−^, the following redox reaction occurred between the exposed electrode surface and the electrolyte.
(1)$${\left[Fe{\left(CN\right)}_{6}\right]}^{3-}+e^{-}\;↔\;{\left[Fe{\left(CN\right)}_{6}\right]}^{4-}$$

Figure 4A,B shows the electrochemical performance of a Au substrate, while Figure 4C,D shows the electrochemical performance of a Au microgap electrode. Current peaks of the redox species [Fe(CN)_6_]^3−^^⁄4−^ were confirmed at 160 and 320 mV in the CV of each process (Figure 4A,C). After immobilization of the MF-3WJ on the substrate, the redox peak current value decreased remarkably. The negative charge of the phosphate groups contained in the DNA backbone interfered with the redox reaction between the electrode surface and the electrolyte, blocking the current. An additional pair of redox peaks were confirmed on the substrate modified with Ag ion. This redox peak occurred when Ag ions intercalated in MF-3WJ exchanged electrons between Ag^+^ and Ag^2+^. EIS is an electrochemical impedance spectroscopy method that has been widely used as an effective and fast method to measure the impedance value of the electrode surface during the frequency change process. In the Nyquist plot, the half-circle diameter refers to the charge transfer resistance (R_ct_) of the working electrode surface. Figure 4B,D depicts the Nyquist plots of a Au microgap electrode and Au substrate. EIS was performed in the same buffer as the CV. When comparing bare and modified electrodes with MF-3WJ, the change in R_ct_ was higher in the modified electrodes. This finding confirmed that MF-3WJ DNA was immobilized on the surface. The negatively charged phosphate backbone of MF-3WJ induced larger R_ct_ because it hindered the movement of [Fe(CN)_6_]^3−^^⁄4−^ to the electrode surface. In the substrate modified with Ag ions, the R_ct_ was markedly reduced. In MF-3WJ, it was confirmed that the intercalation of Ag ions can significantly enhance electron transfer, providing more active regions than bare Au, which is known as a good conductive material.

Each process was verified on Au substrates and Au microgap electrodes through CV and EIS. Table 1 shows the comparison of the electrochemical performance of the Au substrate and the Au microgap electrode. Δ is the difference between the signal in bare and the substrate modified with 3wj+ag. ΔIpc/area and ΔR_ct_/area were calculated as signals generated per unit area of the working electrode. When comparing the electrochemical performances of the Au substrate and the Au microgap electrode, the Au microgap electrode showed superior performance.

### 3.3. Electrochemical Biosensor Performance

To evaluate the performance of the fabricated electrochemical biosensor, SWV was performed in 10 mM HEPES. Different concentrations of IFN-γ (10 ng/mL to 1 pg/mL) were used as samples, and Ag^1/2+^ intercalated into MF-3WJ was used as an electron mediator in SWV. The SWV signal of the MF-3WJ/Au microelectrode cultured with different concentrations of IFN-γ is shown in Figure 5A. This signal was obtained using artificial IFN-γ samples diluted with PBS. As the concentration of IFN-γ increased, the corresponding peak current gradually decreased. This was because more antigen–antibody immune complexes that act as insulating layers were formed to reduce the signal. Figure 5B shows the linear regression curve with an intercept of 8.4 × 10^−7^ ± 1.5 × 10^−7^ and a slope of 1.9 × 10^−7^ ± 1.2 × 10^−8^ (R^2^ = −0.98). The linear regression curve was obtained using the peak current value. The limit of detection (LOD) of the electrochemical biosensor using this regression curve was 0.67 pg/mL. In addition, IFN-γ samples diluted in 10% human serum were prepared, and SWV was performed in the same concentration range to confirm that they could be used in artificial IFN-γ samples and actual clinical samples. Figure 5C shows the SWV signal in clinical IFN-γ samples. It was observed that in clinical IFN-γ samples, the corresponding peak current decreased as the concentration of the samples increased. Figure 5D shows the linear regression curve with an intercept of 9.5 × 10^−7^ ± 7.2 × 10^−8^ and a slope (R^2^ = 0.99) of 1.5 × 10^−7^ ± 7.4 × 10^−9^. The LOD of the electrochemical biosensor using this regression curve was 0.42 pg/mL. This suggested the possibility of detecting IFN-γ in actual clinical samples. To evaluate the manufactured electrochemical biosensor’s selectivity, SWV was performed after reacting different proteins (myoglobin, hemoglobin, BSA, IFN-γ, CRP, TNF-α) at the same concentration of 0.25 μg/mL (Figure 5E). Results from the selectivity experiment showed a remarkably lower signal for IFN-γ compared with other proteins. Therefore, it was confirmed that the fabricated electrochemical biosensor had high selectivity. Table 2 presents a comparison of the performances of IFN-γ sensors reported in the literature and the electrochemical biosensor proposed in this study.

## 4. Conclusions

In this study, we developed an electrochemical DNA aptasensor with a microgap electrode for IFN-γ detection. For minimizing the detection process, the present electrochemical biosensor comprised a multifunctional DNA aptamer as a bioprobe. Moreover, a Au microgap electrode provided minimal sample loading volumes and enabled the seven sample measurements in one electrode. Using the Ag^+^ inserted between MF-3WJs as the redox label, the change in the electrochemical signal of IFN-γ was confirmed through SWV. It was observed that IFN-γ diluted with PBS showed a dynamic range of 1 pg/mL–10 ng/mL, and the LOD was 0.67 pg/mL. IFN-γ diluted with 10% human serum also showed the same dynamic range, and the LOD was 0.42 pg/mL. The proposed electrochemical biosensor has several advantages compared with those reported in previous studies. For example, it could react directly with IFN-γ without the need for multiple labeling and washing processes. In addition, it increases reliability as it can be measured seven times with one electrode. Furthermore, it has a low detection limit and a wide dynamic range. In the future, this electrochemical biosensor platform may be used not only for IFN-γ detection and molecular diagnosis with the low sample volume, but also for the detection of several biomarkers in immunological and cancer research.

## Figures and Tables

**Figure 1 biomedicines-09-00692-f001:**
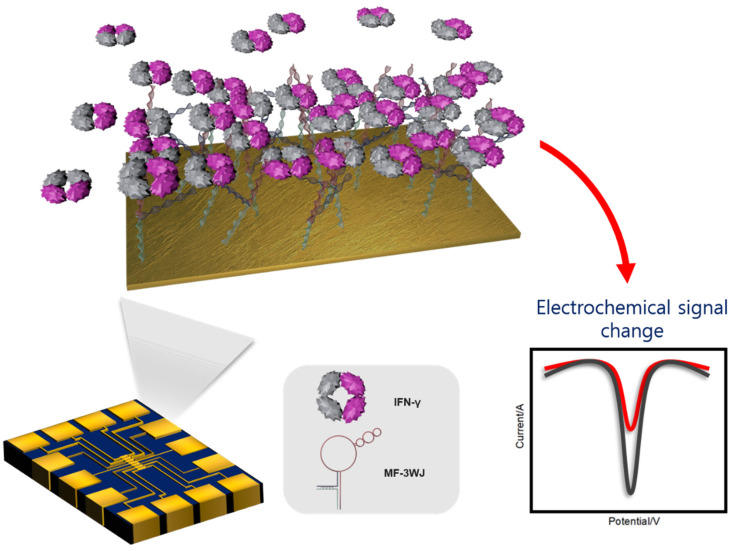
Concept image of the proposed IFN-γ electrochemical biosensor. IFN-γ: interferon gamma, MF-3WJ: multifunctional triple-stranded aptamer, SWV; square wave voltammetry.

**Figure 2 biomedicines-09-00692-f002:**
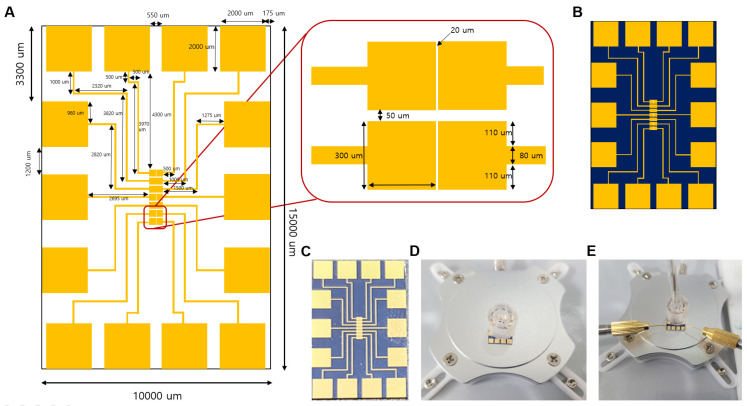
(**A**) Designed image of the Au microgap electrode. (**B**) Image of the Au microgap electrode. (**C**) Photo image of the Au microgap electrode. (**D**) Photo image of the Au microgap electrode with the working chamber. (**E**) Photo image of electrochemical measurement with a probe tip in the working solution.

**Figure 3 biomedicines-09-00692-f003:**
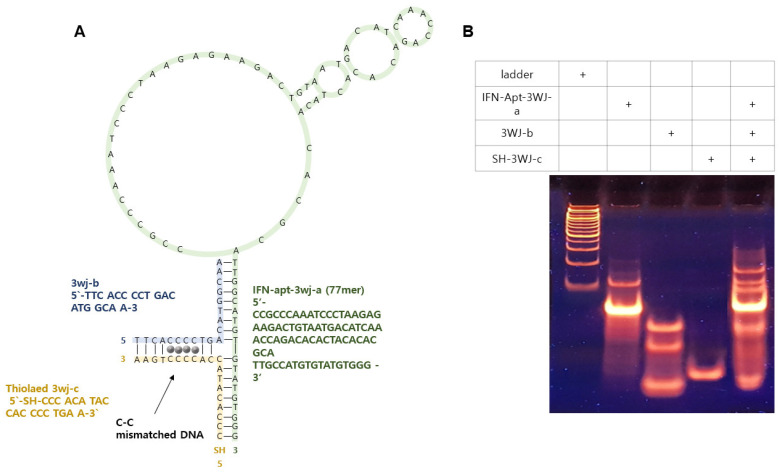
(**A**) Schematic diagram of MF-3WJ predicted by folding. (**B**) TBE PAGE confirmed to DNA 3WJ assembly and DNA ladder (lane 1), IFN-γ-Apt-3wj-a (lane 2), 3wj-b (lane 3), SH-3wj-c (lane 4), and MF-3WJ (lane 5).

**Figure 4 biomedicines-09-00692-f004:**
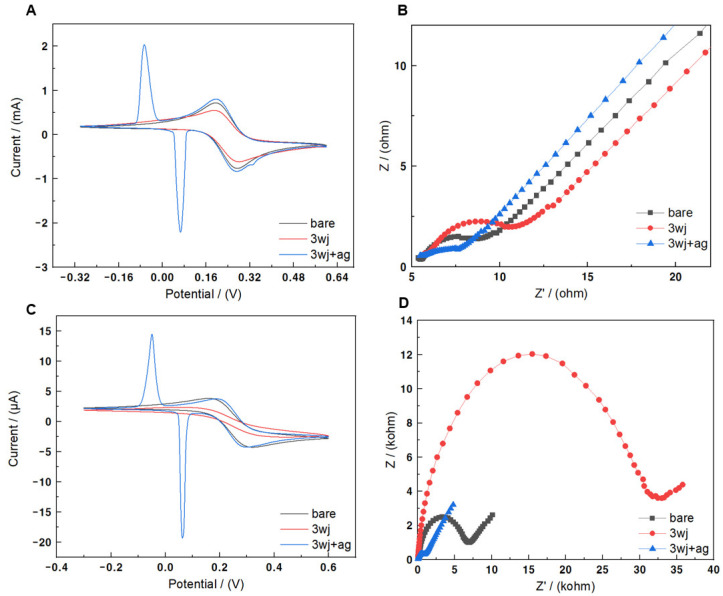
(**A**) Cyclic voltammogram of Au substrate (black line), MF-3WJ on Au substrate (red line), and MF-3WJ intercalated with Ag^+^ on Au substrate (blue line). (**B**) Electrochemical impedance spectra of different modified Au substrates with bare, MF-3WJ, and MF-3WJ/Ag ion. (**C**) Cyclic voltammogram of Au microgap electrode (black line), MF-3WJ on Au microgap electrode (red line), and MF-3WJ intercalated with Ag ions on Au microgap electrode (blue line). (**D**) Electrochemical impedance spectra of different modified Au microgap electrodes with bare, MF-3WJ, and MF-3WJ/Ag ions. Working solution: 10 mM HEPES (pH 7.03) and 5 mM K_3/4_Fe(CN)_6_.

**Figure 5 biomedicines-09-00692-f005:**
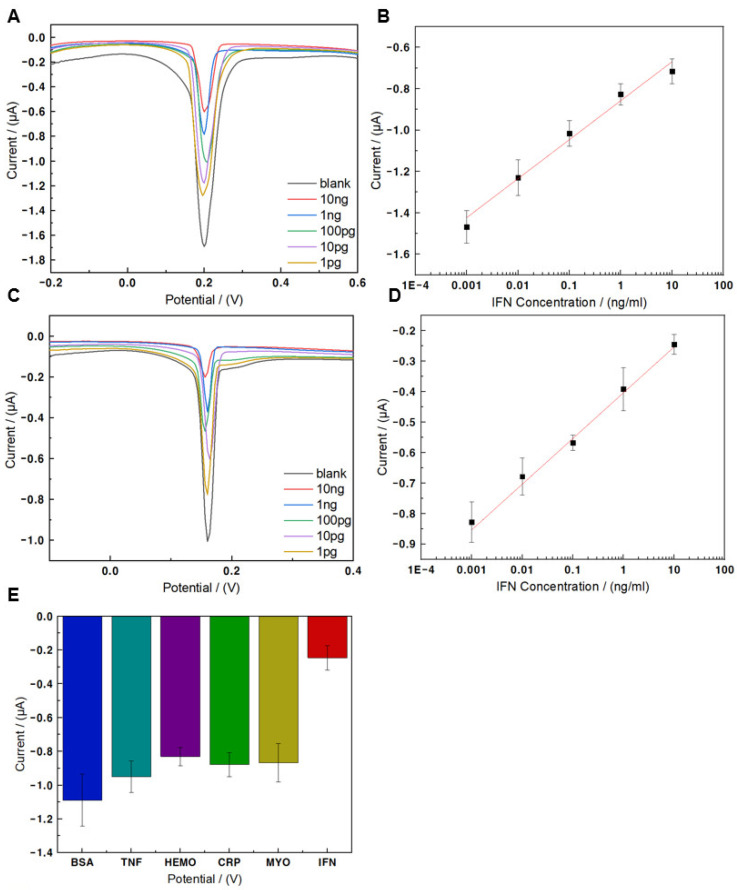
(**A**) Square wave voltammetry at different concentrations of interferon gamma (IFN-γ) diluted with phosphate-buffered saline (PBS) in 10 mM HEPES (pH 7.03). (**B**) Calibration curve of different concentrations of IFN-γ in PBS, linear range from 10 ng/mL to 1 pg/mL. (**C**) Square wave voltammetry at different concentrations of IFN-γ diluted with 10% human serum in 10 mM HEPES (pH 7.03). (**D**) Calibration curve of different concentrations of IFN-γ in 10% human serum, linear range from 10 ng/mL to 1 pg/mL. (**E**) Current of several proteins based on selectivity.

**Table 1 biomedicines-09-00692-t001:** Comparison of the electrochemical performances of the Au substrate and the Au microgap electrode.

	Au Substrate	Au Microgap Electrode
3wj + ag Ipc	2.21 mA	0.019 mA
ΔIpc/area	0.014 mA/m^2^	0.21 A/m^2^
3wj + ag R_ct_	2.74 Ω	890 Ω
ΔR_ct_/area	1.71 × 10^4^ Ω/m^2^	9.89 × 10^9^ Ω/m^2^

**Table 2 biomedicines-09-00692-t002:** Comparison of electrochemical biosensors for IFN-γ detection reported in the literature and the electrochemical biosensor proposed in this study.

Probe	Detection Method	Detection Range	LOD	Ref
Antibody	EIS	5–1000 pg/mL	3.4 pg/mL	[29]
Aptamer	CV	10–1500 pg/mL	3 pg/mL	[30]
Antibody	EIS	0.1–1000 ng/mL	0.1 ng/mL	[31]
Aptamer	DPV	0.01–50 ng/mL	0.0003 ng/mL	[32]
Antibody	SWV	1–500 pg/mL	0.34 pg/mL	[33]
Aptamer	SWV	1 pg/mL–10 ng/mL	0.42 pg/mL	This study

## Data Availability

Not Applicable.

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
