# Peer review of "Fabrication of an Electrochemical Aptasensor Composed of Multifunctional DNA Three-Way Junction on Au Microgap Electrode for Interferon Gamma Detection in Human Serum"

_biomedicines, 2021, doi:10.3390/biomedicines9060692_

Round 1

Reviewer 1 Report

The authors report on a biosensor for interferon gamma detection by making a device based on a gold microgap electroctrode functionalized with three-way junction aptamer structure for the analyte, and its electrochemical characterization. While the exploited readout platform has been reported on previously, including also combined with aptamers to ensure specificity, the main novelty in the present manuscript is the use of another aptamer to target determination of another bioanalyte. Despite that migrating previously reported core strategies by adaption the aptamer sequence to another bioanalyte may be considered incremental, such adoption do have practical values. Although the significance could have been larger, there are other concerns in the presentation that appear to be of more concern. In particular, there appear to be major concerns related to selectivity. The authors should adequately address major concerns as outlined in the following before a final recommendation can be made.

Selectivity: The data presented in Fig 5E raises a few questions. (On the technical side: Why is the x-axis depicted as electrical potential?). It is stated that these data are obtained at the same concentrations – is this mass or molar concentration? If mass: is this the relevant concentration measure to compare at? Does the lower signal for the IFN-gamma indicate that the signal from this cytokine would be “drowned” in smaller variation of other components in a real liquid biopsy?

Sequence of the IFN-gamma aptamer: Although the nucleotide sequences are stated in the manuscript, it is not entirely evident what is the basis for this (more clear citation?).

Reproducibility: What are the uncertainties depicted in Figs. 4B and D? Are these reflecting uncertainties from repeated experiments, including e.g. sensor fabrication? How reproducible is the sensor fabrication? What are the uncertainties in LOD based on data as presented in Figs. 4B and D?

Assembly of MF-3WJ: the authors claim that new bands on the TBE-PAGE (lane 5) indicate the aptamer assembled state – but it also appears that most of the material is not converted to the assembled aptamer. Based on the notion that a lot of the material is not assembled, the authors are encouraged to assess the fraction of material converted and the impact on the non-converted fraction on the functional biosensor.

Number of significant figures: The authors state the parameters in the fitted linear regression curves apparently without considering the significance of the obtained parameters. One example is the statement: ….”intercept of 8.389E-7± 1.50845E-7..” (line 208). A more appropriate statement would be to consider the number of significant figures in view of the uncertainty (e.g. (8.4 ± 1.5)xE-7 ). A facet related to this: what are the uncertainties of the data reported in Table 1?

Some language issues: The first sentence in the introduction appear not to be complete. Some rephrasing on the last section of the introduction could be considered with change to a state that this is summarizing the approach and some major findings (but it is not considered essential to include main results here). There are some statement on concentration as “Mm” where it probably should be “mM”. If the name abbreviated IDT is the company providing the custom designed sequences of some of the aptamer sequences, the full name of the company is probably not the one given. The test concentration range of IFN given in line 201 appear not be stated correctly. The concentration ranges stated in the legend to Fig. 5 appear not to be aligned with the data in the graphs.

Reviewer 2 Report

In this manuscript the authors describe the fabrication of a very interesting electrochemical aptasensor for Interferon gamma (IFN-γ), whose detection they evaluate in both artificial and real samples. The authors use multifunctional triple-stranded aptamers (MF-3WJ) which are designed to contain an IFN-γ aptamer sequence, an anchoring region and a 4C-C (cytosine-cytosine) mismatch sequence where Ag+ ions were intercalated. The reliability of the electrochemical aptasensor was ingeniously shown by measuring seven samples from one Au micro-gap electrode. The aptasensor shows an overall good performance and also a comparison with literature was done.

Overall, the manuscript is well written, innovative and interesting; however, it seems superficial in some parts and requires a more detailed description of the fabrication process and the evaluation steps undertaken by the authors. The reviewer suggests the authors to check the following article: „An amplified label-free electrochemical aptasensor of γ-interferon based on target-induced DNA strand transform of hairpin-to-linear conformation enabling simultaneous capture of redox probe and target - Biosens Bioelectron. 2019 Dec 1;145:111732. doi: 10.1016/j.bios.2019.111732”, especially the experimental section 2, on how detailed the fabrication steps and experimental conditions (optimization) are described.

More specific required corrections:

  • Lines 27-28, replace “It was observed that, it showed linearity” with “A linear region was observed …..”
  • Keyword „biosensor” should be deleted, the authors already mentioned “electrochemical biosensor”
  • Line 60, replace “and is thus is easy to modify” with “which is easy to modify, and ….”
  • Line 62: replace redox mark with redox label, which is used throughout the manuscript
  • Lines70-77 (last paragraph of introduction) should be revised since it is rather part of the “experimental section”. The reviewer would suggest to keep the aim of the work, and move all technical details, including references to sections 2.2 and 2.3. Reference [22] should be under section 2.3, and a short summary of ref [22] must be provided (e.g. on a substrate of SiO2, metal deposition of Cr (2 nm) and Au (50 nm) was carried out).
  • Line 115-116: “For pre-treatment of the electrodes, the electrodes were performed for 15 min with ultrasonication in acetone solution.” Do the authors mean that before measurements, a pre-treatment step consisting of 15 min ultrasonication in acetone is necessary?
  • Line 131 – was the pH (7) value optimized? Why not use physiological pH 7.4? In the figure legends a pH value of 7.03 is given. Please keep the same notation in text and legends.
  • Line 135: correct “−0.3~0.6 V” to “−0.3 to 0.6 V”
  • Line 142: Ag+ ions do indeed improve the electron transfer, but how can the authors be sure that the silver ions stabilized the duplex DNA strand?
  • Line 155: “self-assembly process” – presumably due to the thiol groups? A short comment should be added on how the immobilization occurred.
  • Line 156: provide also area of the Au substrate
  • Line 174: Since the charge transfer resistance is mentioned, the electrical equivalent circuit should be inserted in Fig 4.B and/or D. From the looks of it, the same circuit should apply for both electrodes, and the difference should be given by resistance and capacitance values.
  • Line 182: at the end, the authors should highlight that the presence of Ag ions significantly improves the electron transfer, Rct is lower than for bare Au, where it is known that Au is a good conductive material.
  • Table 1. The authors compare the two substrates using the normalized unit of measure for the two parameters, but the same unit of measure should be used also for the “brute” parameters. Choose either mA or μA, and ohm or kohm, where ohm has the symbol Ω.
  • Lines 195-196 and from the values in the table, the Au substrate shows in the reviewers opinion a better performance compared to the Au micro-gap electrode in terms of Rct values (lower Rct values mean a better electron transfer). Please comment on that.
  • Line 201: “10 ng/mL–1 pg/mL)” – delete “-1”
  • Lines 221-222: “at the same concentration (Figure 5E)”. Which concentration would that be? Please provide more details on how the experiment was conducted. Usually, no signal, or low signal is expected from the interferent when compared to the analyte of interest. If the interferents have different reduction potentials, then that should be highlighted with an overlay SWV spectra for all interferents.

Round 2

Reviewer 1 Report

This is a revised version of a previously submitted manuscript. The concerns raised on the initially submitted manuscript have been adequately addressed by the authors, and publication is recommended.

Reviewer 2 Report

Dear Authors,

Although some more effort/input was expected from your side, the requirements were met/answered.

Thank you.